# Artificial Intelligence Adoption by SMEs to Achieve Sustainable Business Performance: Application of Technology–Organization–Environment Framework

**Saeed Badghish**  **and Yasir Ali Soomro \***

Faculty of Economics and Administration, King Abdulaziz University, Jeddah 21589, Saudi Arabia; sbadghish@kau.edu.sa
* Correspondence: ysomro@kau.edu.sa

**Abstract:** The primary purpose of this study was to investigate and present a theoretical model that identifies the most influential factors affecting the adoption of artificial intelligence (AI) by SMEs to achieve sustainable business performance in Saudi Arabia by integrating the Technology–Organization–Environment (TOE) framework. The authors utilized a quantitative method, using a survey instrument for this research. Data for this research were collected from managers working in six different sectors. Subsequently, based on company size, firms were divided into two groups, allowing multi-group analysis of small and medium-sized businesses to explore group differences. Hence, firm size played a moderating role in the conceptualized model. Data analysis was performed on SmartPLS 3, and the results suggest that dimensions of the TOE framework, such as relative advantage, compatibility, sustainable human capital, market and customer demand, and government support, play a significant role in the adoption of AI. Moreover, this study found a significant influence of AI on SMEs' operational and economic performance. The multi-group analysis (MGA) results reveal significant group differences, with a medium-sized firm strengthening the relationship between relative advantage and AI adoption compared to small-size firms. The findings lead to practical implications for companies on how to increase the adoption of AI to help SMEs embrace their technological challenges in KSA and obtain sustainable business performance to contribute to the economy.

**Keywords:** TOE; AI adoption; sustainable business performance; SMEs



## 1. Introduction

In a rapidly transforming and increasingly digitalized society, interest in artificial intelligence (AI) is growing. Artificial intelligence (AI) has received increasing attention from various areas of society, industry, and business [1]. AI is referred to as the Fourth Industrial Revolution [2]. AI is a field that combines computer science with large datasets to improve the quality of business decision making. Artificial intelligence is the simulation of human intelligence by machines (programs) using technologies such as machine learning, deep learning, data mining, natural language processing, image recognition, and more [3,4]. AI and big data empower people to systematize disaggregated information in a system and transform data into actionable business decisions, thus accelerating company-wide decision making [5–7]. Several studies have examined AI adoption and its influence on business performance by reducing costs and enhancing forecasting [8], improving business operations [9], delivering increased productivity by substituting typical human everyday jobs with automation [10], enhancing product innovation [11], and fostering firm growth [12,13]. Hence, businesses are focusing more on AI, and there is tremendous potential for AI to enhance the performance of firms [14], but there are also major obstacles to the adoption of AI by companies [15].

Researchers in academia consider the influence and implications of AI technology to be the most important research area [16], as acceptance of AI practices also impacts the financial and non-financial performance of SMEs [17]. Studying the mechanisms and key factors of the impact of AI on firm performance has significant theoretical and practical value [18]. Consequently, there is a compelling need to investigate the multidimensional factors (particularly qualitative factors) that influence the adoption of AI within SMEs.

Small and medium-sized enterprises (SMEs) are an essential driver of economic development [19] and are essential for most economies around the world, especially in developing and emerging countries [20]. Unlike large companies, SMEs are highly resilient to technical change and have better adaptability to market fluctuations, while making fast decisions due to their organizational structure [21,22]. SMEs use disruptive technologies to expand their businesses and advance their operational activities [23]. The current industrial revolution has driven up the demand for SMEs to adopt digital technology [24,25]. Due to pressure from stakeholders, small and medium-sized enterprises (SMEs) have recently begun to take on innovation initiatives [19]. Saudi Arabia is a leading oil-producing country in the world [26] and is experiencing rapid industrial and economic growth. The Ministry of Labor and Social Development estimates that Saudi Arabian SMEs contribute approximately 22 percent of the kingdom's gross domestic product. Approximately 34% of Saudi workers were employed by small and medium-sized enterprises (SMEs) in 2019 [21]. Saudi Arabia has implemented Vision 2030, which is a strategic document to foster growth and encourage innovation adoption in each sector of the country; hence, SMEs are supported to adopt new technologies and environmentally friendly production processes [27,28]. A previous study in the context of Saudi Arabian SMEs and AI by Baabdullah et al. (2021) laid the foundation by investigating the antecedents and consequences of AI practices within B2B firms and calling for more research into AI adoption [17]. Moreover, a study utilized the integrated technology acceptance model (TAM)–TOE model to understand factors influencing AI adoption within firms [29]. Hence, to add to the literature, the present study utilizes the Technology–Organization–Environment (TOE) framework to construct a research model that explains the readiness of firms towards the adoption of AI and the firm performance within SMEs in emerging countries like Saudi Arabia. Secondly, most previous studies tend to treat performance as a one-dimensional construct [30,31]. In this study, the authors conceptualize performance as a two-dimensional construct (i.e., operational and economic performance). To the authors' knowledge, this study is the first research work focusing on SMEs' AI adoption and its impact on performance as a two-dimensional construct (i.e., operational and economic performance) in Saudi Arabia. Lastly, firm size was used as a moderator variable to understand the group differences between small and medium-sized SMEs.

## 2. Literature Review

### 2.1. Technology–Organization–Environment (TOE) Framework

In organizational studies on innovation diffusion and adoption, two widely utilized theories are the diffusion of innovation (DOI) theory [32] and the Technology–Organization–Environment (TOE) framework [33]. This research excludes other prevalent theories like the technology acceptance model (TAM) [34], the Theory of Planned Behavior (TPB) [35], and the Unified Theory of Acceptance and Use of Technology (UTAUT) [36] as they primarily address individual decision-making processes. This study only utilizes the TOE framework. The TOE framework [33] provides a comprehensive framework for analyzing the interplay between technology characteristics, organizational factors, and environmental influences in shaping the adoption decisions of organizations. The TOE framework offers a theoretical perspective for analyzing the adoption of innovative technology within organizations [37]. TOE is a classical framework that suggests general factors that explain and predict innovation and technology adoption probability [33]. The framework proposes three business context components that impact the adoption and implementation of

technological advances. Technology development [38], organizational conditions [39], and industry environment [40] make up the TOE framework.

The TOE framework has been utilized in many prior studies in various contexts of technology. Scupola, in his study, focused on understanding and presenting a comprehensive analysis of the adoption of Internet commerce by SMEs utilizing the TOE framework, considering various external and internal factors tailored to the unique context of southern Italy [41]. Furthermore, researchers investigated the determinants of cloud computing adoption in the manufacturing and services sectors, offering insights into the factors influencing organizations' decisions to adopt cloud technologies [42]. In addition, an earlier work also shows that TOE framework constructs are more likely to apply to large companies [43]. Extending the TOE framework, studies have employed it to explain the adoption of specific technological innovations. For instance, the model has been applied to understand the adoption of electronic customer relationship management (eCRM) systems [44], cloud computing [45,46], geographical information technologies [47], blockchain technology [48,49], e-business [50], green banking practice [51], and other emerging technologies.

A recent study has attempted to integrate TAM and TOE to provide a robust framework for analyzing and enhancing the adoption of AI technologies in construction firms [52]. In addition, an extended TOE framework has been empirically tested to analyze the utilization of online retailing in the digital transformation of the Vietnamese business landscape [53]. Another study in the context of AI [42] conducted a comparative case study to explore the adoption of artificial intelligence in public organizations, providing insights into the factors and dynamics shaping AI adoption in the public sector. Another study has investigated the antecedents of MLOps (machine learning operations) adoption, employing the TOE framework for analysis [54].

Recently, a few studies have investigated AI adoption in the context of SMEs to examine AI technology's applicability in different situations such as AI-based Business-to-Business (B2B) practices [17] and accounting automation [24]. Moreover, a study examined the determinants of performance in the adoption of artificial intelligence within the hospitality industry during the COVID-19 pandemic, shedding light on factors influencing the successful integration of AI technologies utilizing the TOE framework [55]. A foundational study conducted in the realm of Saudi Arabian SMEs and AI explored the precursors and outcomes of AI practices in B2B firms [17], prompting further investigation into AI adoption. Additionally, another study employed the integrated technology acceptance model (TAM)–TOE model to comprehend the factors shaping AI adoption within organizations [29]. This article makes a dual contribution. Firstly, it delves into both the direct and indirect impacts of TOE characteristics on AI adoption. Secondly, it provides a more comprehensive evaluation of the factors influencing AI adoption compared to previous studies and further investigates how AI adoption decisions will affect firm performance in the context of Saudi SMEs.

## *2.2. Hypothesis Development*

### 2.2.1. Technological Readiness

Technological readiness represents the set of technologies available to a company [41]. Technological features can be viewed as cognitive beliefs of workers reflected in attitudes towards technological innovation and can influence the adoption of innovation [56]. Various technological factors that include comparative advantage, compatibility, complexity, testability, traceability, ease of use, perceived usefulness, the intensity of information, and uncertainty have been investigated [24,56,57]. The focus of this study is cost, relative advantage, compatibility, and complexity.

Cost is seen as one of the main obstacles to adopting technologies [58,59]. The costs include implementation costs, such as the financial and human resources required for the implementation [33]. However, it is believed that high implementation costs could motivate innovation users to take innovations seriously and employ them proactively to make innovation adoption more cost-effective [60]. The higher investment to adopt AI

may be an obstacle and may result in firms' reluctance to simplify interpretations and heightened sensitivity to errors and unusual occurrences, regardless of their size. Thus, the following hypothesis was formulated:

**H1(a):** *Implementation cost has a negative effect on AI adoption.*

Relative advantage is defined as the extent to which an innovation is considered better than its predecessors [32]. Organizations are more likely to choose technology that offers better performance and higher financial returns than other technologies [60]. Relative advantage is positively related to innovation adoption [61]; the greater the relative advantage, the faster the adoption of innovation. Studies have shown that relative advantage is positively related to the adoption of innovation [62,63]. For SMEs, AI produces several relative benefits: reduced costs, quick decision making, and forecasting. In a highly competitive market, these benefits are essential for companies. Further, a study indicates that relative advantage strongly influences AI adoption [64]. Hence, the following hypothesis was formulated:

**H1(b):** *Relative advantage has a positive effect on AI adoption.*

Complexity is the extent to which an innovation is considered relatively difficult to understand and use [60]. Technological complexity refers to difficulties in learning to study and understand new technologies [65], and it is commonly believed that complexity has a negative impact on innovation adoption [66]. In addition, a study indicates that excessive technological complexity decreases the ability to perform competently [67]. The existing literature on the diffusion of innovations has shown that the acceptance rate decreases with the increasing complexity of implementing an innovation [68,69]. AI can be challenging [64], yet different applications have different levels of complexity, making it interesting to study. In the context of AI, if the managers find the AI technology to be complicated to implement and understand, they will likely avoid the AI adoption within firm. Therefore, we hypothesize:

**H1(c):** *Complexity has a negative effect on AI adoption.*

Compatibility is the degree to which innovation is seen as compatible with a company's existing values, experience, and needs [70]. Technology compatibility is a necessary requirement: when green innovation requires resources that are not available to the organization or creates change that is not in line with its strategy, implementation becomes very difficult [37]. Previous studies suggest that compatibility positively influences innovation adoption [37,61,71]. A similar notion will be observed when adopting AI and its compatibility. If SMEs believe AI technology meets all their work requirements and innovation prerequisites, they are going to be more receptive to implementing it [24]. Therefore, we postulated the following hypotheses.

**H1(d):** *Compatibility has a positive effect on AI adoption.*

2.2.2. Organizational Readiness

Organizational readiness refers to the distinctiveness, structures, processes, and resources that limit or facilitate the adoption of technological innovation [72]. A number of organizational variables, such as the quality of human resources, top manager leadership skills, organizational support, and organizational culture, have been discussed in relation to green innovation adoption [73,74]. Previous studies have focused primarily on organizational support and sustainable human capital, as these factors have consistently proven to be more important in influencing green innovation adoption [75,76]. Hence, this study

adopts the same variable in the organizational context to check how these factors influence AI adoption.

Organizational support refers to the extent to which a company helps employees use a particular technology or system [77]. Stimulating innovation and ensuring the availability of financial and technical resources for new technology or innovation positively affect the uptake of innovation [78]. Top management plays a vital role in providing organizational support. To ensure the successful implementation of AI initiatives, top management's main task is to acquire resources and allocate them efficiently, so that companies can achieve a competitive advantage [79,80]. Therefore, the authors postulated the following hypotheses:

**H2(a):** *Organizational support has a positive effect on AI adoption.*

Sustainable human resource management is the combination of the sustainability concept with HR [81] that enables firms to have sustainable human capital. In response to the dynamic nature of the business environment, firms are actively seeking to include artificial intelligence (AI) as a catalyst for significant change. The role of the human resources department in facilitating the change becomes pivotal for industry 4.0 adoption [82]. Many studies highlight the importance of sustainable human resource (SHR) policies in shaping an organization's ability and readiness to embrace artificial intelligence (AI), as AI-based systems will significantly change the nature of the workforce [83,84]. Concurrently, there has been an increased emphasis on the adoption of sustainable human resource (HR) practices by organizations [8] as they endeavor to harmonize their activities with principles of environmental, social, and economic sustainability [81]. Sustainable human resources (HR) practices involve a diverse array of techniques and activities that are designed to cultivate a work environment conducive to the long-term promotion of well-being, equity, and organizational resilience to new advancements, and to reducing the skill gap through employee development initiatives and training and rewards aimed at fostering work–life equilibrium and employee well-being, for the implementation of new digital technologies [8,81,85]. The increasing prevalence of AI technology and the several benefits outlined have led to a corresponding rise in the need for skills connected to AI [86]. The implementation of sustainable human resources (HR) strategies that emphasize employee development and upskilling can effectively provide the workforce with the essential skills required for the effective application of artificial intelligence (AI). Organizations may effectively equip their human capital to adapt to the ever-changing technology landscape by making investments in ongoing learning and development initiatives, to have sustainable human capital to obtain long-term sustainability [87]. The presence of skilled human capital and the dedication of managers to learn and embrace new technology are imperative and seen as one of the main factors that motivate companies to invest in innovative technology [88]. Sustainable human capital can minimize uncertainty, tolerate risk, and reduce resistance to innovation and green practices [89]. However, the literature on sustainable human capital is still limited [90], especially in the context of AI. Therefore, the authors expect that having sustainable human capital will positively affect AI adoption and postulate the following hypothesis:

**H2(b):** *Sustainable human capital has a positive effect on AI adoption.*

### 2.2.3. Environmental Readiness

Environmental readiness factors refer to the external pressures that cause a company to pursue technological innovation [37]. Various environmental variables, such as market and customer pressure, stakeholder pressure, government role, and environmental regulation, have been discussed in the literature [74,76]. This study mainly focuses on market and customer support and government support in the context of AI adoption.

Companies need to understand their target customers and anticipate changing preferences, in order to be able to react quickly to market demand and gain a competitive

advantage [91]. Customers are the end-users of the product, so customer demand may have a more significant influence than any other factor in encouraging manufacturers to innovate [92]. Prior studies suggest that markets and customers encourage companies to focus on adopting new innovations [74,76,92]. Therefore, this study proposed the following hypothesis.

**H3(a):** *Market and customer demand has a positive effect on AI adoption.*

Government support refers to assistance or facilitating conditions provided to the employees or organization to transform or implement technology diffusion within the firm. Facilitating conditions are the individual's perception that there is the necessary technical and organizational capacity and infrastructure for them to be able to successfully use new technologies [36]. This factor has been used in various studies that have investigated usage preference or the continued usage of new technology [93,94]. Governments facilitate through monetary incentives or government subsidies and making credit available from commercial banks for implementing technological innovations and encouraging SMEs [95]. Therefore, due to resource constraints, SMEs need additional resources and government support (GS). Therefore, the authors postulate the following hypothesis.

**H3(b):** *Government support has a positive effect on AI adoption.*

### 2.2.4. Artifical Intelligence Adoption and Sustainable Business Performance

AI adoption reduces expenses and improves forecasts and business operations [8,9]. Thus, organizations are focusing increasingly on AI, which has great potential to improve performance [14]. As AI adoption affects SME financial and non-financial performance [17], examining the AI implementation process and key determinants on business performance holds both theoretical and practical significance [18]. The Internet of Things (IoT), artificial intelligence powered by big data analytics, and green innovations are just a few examples of Industry 4.0 elements that can improve sustainable business performance [96]. Recently, performance has been classified into three primary components, which are environmental, economic, and operational [97]. This study has only included two dimensions of sustainable business performance, namely, operational and economic performance, though a few studies have mentioned three dimensions from the perspective of SMEs, as they are critical for sustainable innovation and business performance [21].

Operational performance is determined by a combination of efficient product development, process development, quality compliance, and short lead times [98]. A company's ability to apply AI practices improves certain aspects of its operations and helps in achieving efficiency (e.g., cost reduction, elimination of a liability, etc.). Moreover, AI adoption increases productivity by the automation of tasks [10] and enhances product innovation [11], which in turn improves competitiveness [97]. These innovative measures can improve the quality of operations and reduce production costs by raising the production process's efficiency and lowering input and waste disposal costs. Therefore, this study proposes the following hypothesis:

**H4(a):** *AI adoption positively affects operational performance.*

The economic performance dimension refers to a firm's financial health and growth. Firms that adopt AI technology can experience an increase in revenue, as AI can help firms to better understand their customers and make more accurate predictions about their behavior. AI adoption has clearly reduced the cost of production and allowed for efficiency in decision making that adds to increase profitability and improved financial performance [5,17]. According to a study by the McKinsey Global Institute, AI adoption can increase global GDP by up to 1.2% annually by 2030 [99]. Furthermore, AI can help

firms to reduce costs by automating repetitive tasks and optimizing processes [10,100]. Therefore, this study proposed the following hypothesis.

**H4(b):** *AI adoption positively affects economic performance.*

### 2.2.5. Firm Size as Moderator

Some literature has been written about the determinants of innovation and, in particular, about the influence of company size on innovation [21]. Company size is repeatedly viewed as a relevant organizational characteristic that influences innovation adoption. For instance, the positive influence of firm size was revealed on e-maintenance readiness [75], consistent with these findings. But another study finds that firm size has little effect on technology adoption [101]. This variability in quantifying the impact of firm size could be accounted for by factors such as the type of technology adopted and how firm size is measured [75]. In general, large companies tend to adopt environmentally friendly practices more quickly than small companies because they have adequate resources and vital infrastructure [102].

Recent empirical studies have shown that small companies are much less innovative than large companies [101,103–105]. The reason for this is that, even though large companies may benefit from technological and learning economies of scale, organizational differences in size may outweigh these [106,107]. As a result, small businesses are more likely to be subject to resource and material constraints in innovation than large companies, while large companies are more likely to be subject to behavioral constraints in innovation. In this study, within the SME's context, small and medium firm-size as an organizational characteristic has been taken as a moderator to check whether there are significant differences and influences on AI adoption and firm performance. Hence, the following moderation hypotheses were created:

**H5(a):** *The strength of the association between cost and AI adoption will be stronger and significant in medium-sized SMEs compared to small-sized Saudi SMEs.*

**H5(b):** *The strength of the association between relative advantage and AI adoption will be stronger and significant in medium-sized SMEs compared to small-sized Saudi SMEs.*

**H5(c):** *The strength of the association between complexity and AI adoption will be stronger and significant in medium-sized SMEs compared to small-sized Saudi SMEs.*

**H5(d):** *The strength of the association between compatibility and AI adoption will be stronger and significant in medium-sized SMEs compared to small-sized Saudi SMEs.*

**H6(a):** *The strength of the association between organizational support and AI adoption will be stronger and significant in medium-sized SMEs compared to small-sized Saudi SMEs.*

**H6(b):** *The strength of the association between sustainable human capital and AI adoption will be stronger and significant in medium-sized SMEs compared to small-sized Saudi SMEs.*

**H7(a):** *The strength of the association between market/customer demand and AI adoption will be stronger and significant in medium-sized SMEs compared to small-sized Saudi SMEs.*

**H7(b):** *The strength of the association between government support and AI adoption will be stronger and significant in medium-sized SMEs compared to small-sized Saudi SMEs.*

**H8(a):** *The strength of the association between AI adoption and operational performance will be stronger and significant in medium-sized SMEs compared to small-sized Saudi SMEs.*

**H8(b):** *The strength of the association between AI adoption and economic performance will be stronger and significant in medium-sized SMEs compared to small-sized Saudi SMEs.*

Hence, this research focuses on SMEs' AI adoption and its impact on performance as a two-dimensional construct (i.e., operational and economic performance) in Saudi Ara-bia. Moreover, firm size has been used as a moderator variable to understand the group differences between small and medium-sized SMEs. The conceptualized model is shown in Figure 1.

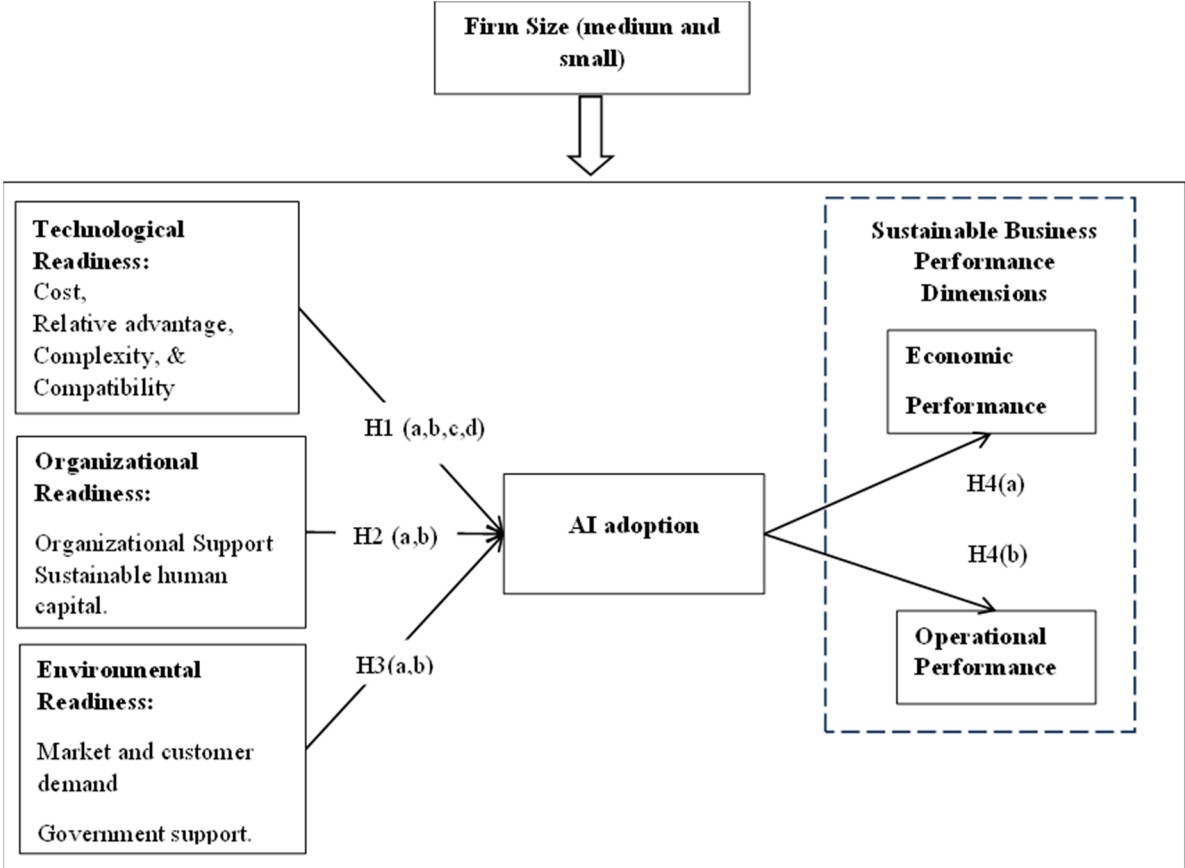

**Figure 1.** Proposed research model.

## 3. Methodology

To provide an empirical interpretation of the conceptual hypothesis developed and presented within the framework, a primary research design employing a quantitative method was employed. Saudi Arabia was selected for conducting this research. Saudi Arabia, one of the world's top twenty economies and the largest in the Arab world and MENA region, is implementing rapid modernization initiatives to achieve its Vision 2030. This dynamic environment offers an intriguing context for investigating AI adoption in small and medium-sized enterprises (SMEs), expected to provide valuable insights for poli-cymakers and practitioners. To determine the primary factors driving the implementation of artificial intelligence (AI) in small and medium-sized enterprise (SME) manufacturing enterprises located in the Jeddah industrial area of Saudi Arabia, a series of semi-structured interviews were performed with five managers. To finalize the questionnaire, the initial step involved identifying the key constructs based on a thorough review of the relevant literature on innovation adoption. Constructs were selected to comprehensively cover the technological, organizational, and environmental factors influencing AI adoption. In the context of our research, we customized the TOE framework originally proposed [33], as elaborated in Section 2. Our approach involved selecting the variables within the

Technology–Organization–Environment (TOE) framework that are most pertinent to Saudi Arabian SMEs. This selection process was informed by interviews with five managers. Each manager was provided with a compilation of contextual variables pertinent to the implementation of artificial intelligence (AI), as identified through a thorough AI studies literature review. Consequently, each manager was tasked with discerning and selecting variables that they deemed relevant to their respective company and industry within the Saudi Arabian context. The criteria for selection stipulated that only variables acquiring 60% or more consensus in favor would be selected in the model. Table 1 presents the summarized outcomes of interviews conducted with five managers.

**Table 1.** Interview results of managers.

| Construct | Sub-Constructs | R-1 | R-2 | R-3 | R-4 | R-5 | Percentage Selection | Variable Selection |
|---|---|---|---|---|---|---|---|---|
| **Technological Factors** | Cost | Yes | Yes | Yes | Yes | Yes | 100% | Selected |
| | Complexity | Yes | No | Yes | Yes | Yes | 80% | Selected |
| | Compatibility | Yes | Yes | Yes | Yes | Yes | 100% | Selected |
| | Relative advantage | Yes | No | No | Yes | Yes | 60% | Selected |
| | Perceived trust | No | No | No | No | Yes | 20% | Not Selected |
| **Organizational Factors** | Sustainable human capital | Yes | Yes | Yes | Yes | Yes | 100% | Selected |
| | Organizational support | Yes | Yes | Yes | Yes | Yes | 100% | Selected |
| | Firm size | No | No | No | No | No | 0% | Not Selected |
| | Organizational policies | No | No | No | Yes | Yes | 40% | Not Selected |
| **Environmental Factors:** | Government support | Yes | Yes | Yes | Yes | Yes | 100% | Selected |
| | Industrial characteristics | No | No | No | Yes | No | 20% | Not Selected |
| | Market and customer factors | No | No | Yes | Yes | Yes | 60% | Selected |
| | Government policies | No | No | No | No | No | 0% | Not Selected |

Finally, the technology factor had four sub-constructs, including cost, relative advantage, complexity, and compatibility [43,61,108–110]. Two sub-constructs were included for the organizational factor: sustainable human capital [75,111] and organizational support [74]. The environmental factor was measured by two sub-constructs, including government support and market and customer factors; each sub-construct contained four items adapted from a previous study [74]. AI adoption (AIA) items were modified in the questionnaire and adopted from previous studies [75,112]. Lastly, sustainable business performance consisted of two distinct components: economic performance with two items and operational performance with two items. All measures of firm performance were modified and adapted from prior research [21,37,113], and all the questionnaire items were finalized for survey (refer Appendix A). A self-administered survey method with random sampling was adopted to obtain data for this study. Random sampling is considered the most suitable strategy due to the equal probability assigned to each unit [114]. Prospective respondents were identified through a systematic approach, leveraging industry directories, business associations, and government records. The finalized questionnaire was distributed to SME executives at middle or senior level or the owner/entrepreneur from the construction, energy, logistics, manufacturing, and services industries in the Jeddah industrial area from

March 2023 till May 2023. Approximately 300 respondents were approached and 220 valid responses from the industry were processed for data analysis. In Saudi Arabia, SMEs are companies with 250 or fewer employees and an annual revenue of less than SAR 200 million (USD 53.3 million). According to data from the Saudi Nitaqat and the General Authority for Statistics (GaStat), the authors classify businesses as either small (6–49 employees) or medium (50–249 employees). Among the 220 responses, 115 were from medium-sized businesses and 105 were from small businesses. Table 2 displays the demographic analysis of the data collected.

**Table 2.** Demographic analysis of respondents.

| Demographics | | Frequency | Percent |
|---|---|---|---|
| Gender | Male | 150 | 68.2 |
| | Female | 70 | 31.8 |
| Job Level | Entrepreneur | 60 | 27.3 |
| | Middle management | 78 | 35.4 |
| | Senior management | 82 | 37.3 |
| Firm Size | Medium (50–249 employees) | 115 | 52.3 |
| | Small (6–49 employees) | 105 | 47.7 |

*Data Reliability*

Table 3 presents the results of a reliability analysis and the measuring model demonstrated strong convergent validity. A number larger than 0.5 for the average variance extracted suggests a high level of validity for both the variable and construct. The loading values of items should fall within the range of 0.05 and 0.07. It was noted that a single item, namely sustainable human capital SHC3, had a low value and was therefore removed from the analysis. The assessment of convergent validity was conducted within three main conditions: The values of the standardized factor loads were found to be more than 0.5. The study found that the composite reliability (C.R) measure was greater than the average variance extracted (AVE) measure. Additionally, the AVE measure surpassed the threshold of 0.5, as recommended [115]. Consequently, a mere 28 elements within the complete model have factor loadings exceeding the threshold of 0.55. Please refer to Table 3 and Figure 2 for further details. The results presented in Table 3 demonstrate a high degree of convergent validity.

**Table 3.** Reliability statistics.

| Construct | Items | Loadings | Cronbach's Alpha (CA) | Composite Reliability (C.R) | Average Variance Extracted (AVE) |
|---|---|---|---|---|---|
| **Technological readiness factors** | | | | | |
| Cost | CTF1 | 0.873 | 0.698 | 0.869 | 0.768 |
| | CTF2 | 0.879 | | | |
| Relative advantage | RTF1 | 0.836 | 0.700 | 0.867 | 0.766 |
| | RTF2 | 0.913 | | | |
| Complexity | XTF1 | 0.924 | 0.830 | 0.922 | 0.855 |
| | XTF2 | 0.925 | | | |
| Compatibility | TFC1 | 0.883 | 0.519 | 0.802 | 0.670 |
| | TFC2 | 0.749 | | | |

**Table 3.** *Cont.*

| Construct | Items | Loadings | Cronbach's Alpha (CA) | Composite Reliability (C.R) | Average Variance Extracted (AVE) |
|---|---|---|---|---|---|
| **Organizational readiness factors** | | | | | |
| Sustainable human capital | SHC1 | 0.845 | 0.661 | 0.794 | 0.504 |
| | SHC2 | 0.841 | | | |
| | SHC3 | 0.477 * | | | |
| | SHC4 | 0.605 | | | |
| Organizational support | OS1 | 0.881 | 0.871 | 0.921 | 0.795 |
| | OS2 | 0.892 | | | |
| | OS3 | 0.902 | | | |
| **Environmental readiness factors** | | | | | |
| Government support | GS1 | 0.751 | 0.848 | 0.897 | 0.686 |
| | GS2 | 0.849 | | | |
| | GS3 | 0.865 | | | |
| | GS4 | 0.843 | | | |
| Market and customer demand | MC1 | 0.823 | 0.806 | 0.886 | 0.722 |
| | MC2 | 0.907 | | | |
| | MC3 | 0.815 | | | |
| **Artificial intelligence adoption** | | | | | |
| AI adoption | AIA1 | 0.746 | 0.727 | 0.847 | 0.648 |
| | AIA2 | 0.841 | | | |
| | AIA3 | 0.826 | | | |
| **Firm performance factors** | | | | | |
| Economic performance | ECP1 | 0.883 | 0.678 | 0.861 | 0.756 |
| | ECP2 | 0.856 | | | |
| Operational performance | OP1 | 0.848 | 0.696 | 0.867 | 0.765 |
| | OP2 | 0.901 | | | |

Note: * Item removed.

According to Fornell and Larcker, in order to establish discriminant validity, it is necessary for the square root values of the average variance extracted (AVE) to be greater than the correlation coefficients between AVE and other variables [116]. Based on the model employed in this study, the authors initially conducted a comparison between the square root of the average variance extracted (AVE) for each construct and the shared variance between the constructs. The authors determined that the square root of the AVE outperformed the shared variance between the constructs. Consequently, the authors are able to assert that there is satisfactory discriminant validity between the constructs, thereby enabling further analysis. Furthermore, the validity of the discriminant is assured, as

evidenced by the fact that the square root of the average variance extracted (AVE) for each measure, as presented in Table 4, exceeds its correlation coefficients with other constructs.

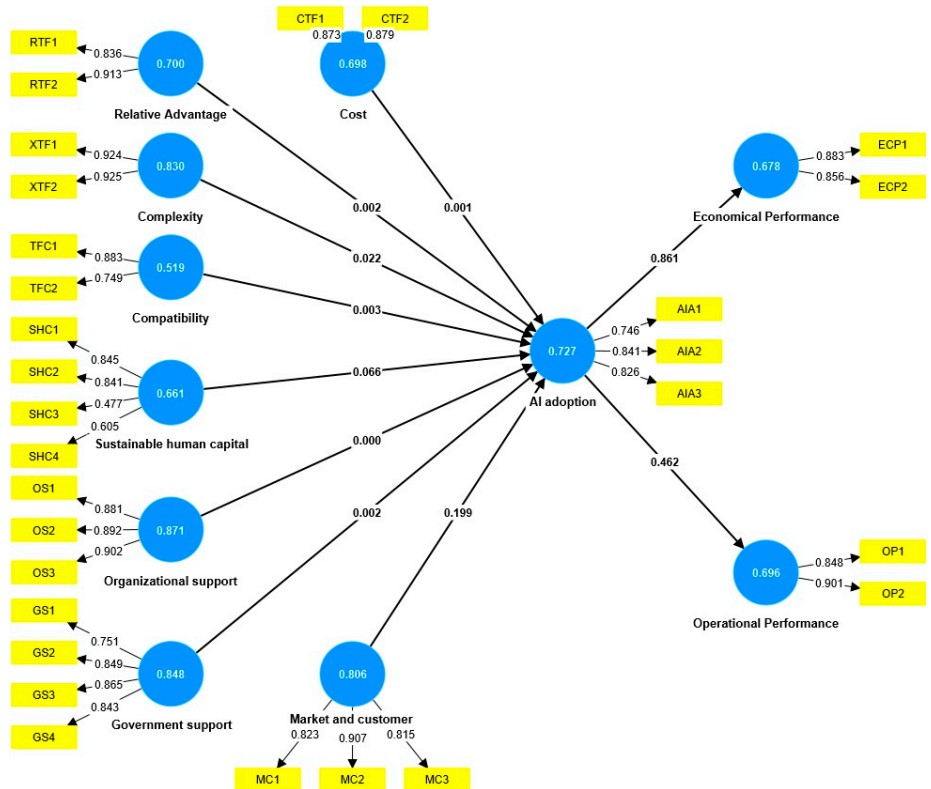

**Figure 2.** Reliability test.

**Table 4.** Co-relation matrix.

|  | AIA | TFC | XTF | CTF | ECP | GS | MC | OP | OS | ECP | SHC |
|---|---|---|---|---|---|---|---|---|---|---|---|
| AI adoption | 0.805 |  |  |  |  |  |  |  |  |  |  |
| Compatibility | 0.551 | 0.819 |  |  |  |  |  |  |  |  |  |
| Complexity | 0.554 | 0.695 | 0.924 |  |  |  |  |  |  |  |  |
| Cost | 0.523 | 0.541 | 0.626 | 0.876 |  |  |  |  |  |  |  |
| Economic performance | 0.680 | 0.445 | 0.468 | 0.326 | 0.870 |  |  |  |  |  |  |
| Government support | 0.565 | 0.481 | 0.414 | 0.529 | 0.297 | 0.828 |  |  |  |  |  |
| Market and customer | 0.724 | 0.581 | 0.540 | 0.599 | 0.458 | 0.709 | 0.849 |  |  |  |  |
| Operational performance | 0.562 | 0.412 | 0.367 | 0.416 | 0.505 | 0.427 | 0.487 | 0.875 |  |  |  |
| Organizational support | 0.362 | 0.336 | 0.213 | 0.345 | 0.132 | 0.592 | 0.440 | 0.394 | 0.891 |  |  |
| Relative advantage | 0.444 | 0.606 | 0.413 | 0.480 | 0.220 | 0.586 | 0.536 | 0.401 | 0.533 | 0.875 |  |
| Sustainable human capital | 0.571 | 0.490 | 0.426 | 0.386 | 0.440 | 0.505 | 0.543 | 0.417 | 0.422 | 0.449 | 0.710 |

Note: TFC = compatibility, XTF = complexity, CTF = cost, ECP = economic performance, GS = government support, AIA = artificial intelligence adoption, MC = market and customer, OP = operational performance, OS = organizational support, RTF = relative advantage, SHC = sustainable human capital.

## 4. Results

PLS-SEM analysis was chosen, in response to the detection of multicollinearity during the reliability test among the variables, making it a suitable approach to address this issue, as recommended [115]. The section on PLS-SEM analysis covers the assessment of validity measures and hypothesis testing (see Table 5 and Figure 3 for reference).

**Table 5.** Hypothetical relationships.

| H.NO | Relationships | Beta | Stdv | T-Value | *p*-Value | Findings |
|---|---|---|---|---|---|---|
| H1(a) | Cost -> AI adoption | 0.042 | 0.050 | 0.832 | 0.204 | Not Supported |
| H1(b) | Relative advantage -> AI adoption | 0.127 | 0.061 | 2.071 | 0.020 *** | Supported |
| H1(c) | Complexity -> AI adoption | −0.036 | 0.041 | 0.865 | 0.195 | Not Supported |
| H1(d) | Compatibility -> AI adoption | 0.090 | 0.058 | 1.554 | 0.002 ** | Supported |
| H2(a) | SHC -> AI adoption | 0.223 | 0.038 | 5.897 | 0.000 * | Supported |
| H2(b) | OS -> AI adoption | −0.025 | 0.041 | 0.625 | 0.267 | Not Supported |
| H3(a) | GS -> AI adoption | 0.080 | 0.051 | 1.566 | 0.006 ** | Supported |
| H3(b) | MC -> AI adoption | 0.432 | 0.055 | 7.892 | 0.000 * | Supported |
| H4(a) | AI adoption -> ECP | 0.672 | 0.034 | 19.643 | 0.000 * | Supported |
| H4(b) | AI adoption -> OP | 0.557 | 0.039 | 14.250 | 0.000 * | Supported |

Note: ECP = economic performance, GS = government support, AIA = artificial intelligence adoption, MC = market and customer, OP = operational performance, OS = organizational support, SCH = sustainable human capital. Significance level at * $p < 0.001$, ** $p < 0.01$., *** $p < 0.05$.

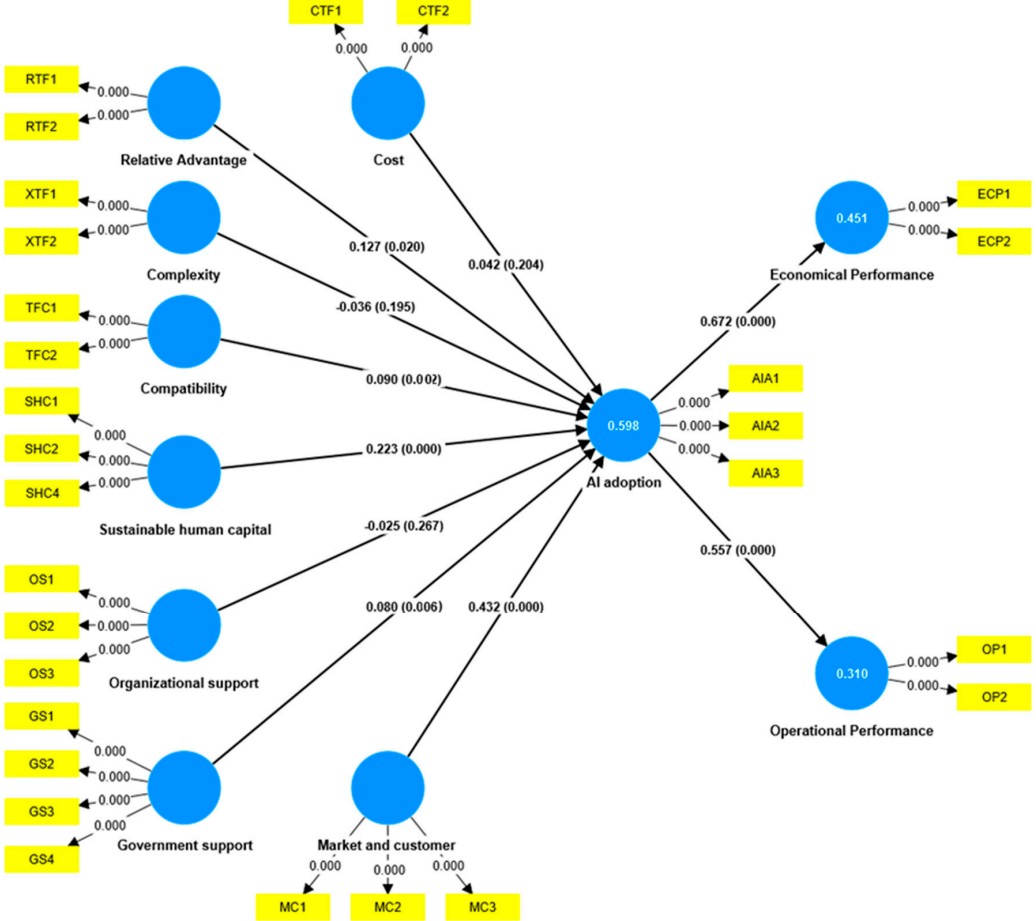

**Figure 3.** Measurement model.

## 4.1. Structural Model Assessment

The structural equation model was used to test the hypothesis, and the results are shown in (Table 5). For technology factors, cost ($\beta = 0.042$, t = 0.832, $p = 0.204$) and complexity ($\beta = -0.036$ t = 0.041, $p = 0.195$) do not have impact on AI adoption while relative advantage ($\beta = 0.127$ t = 0.061, $p = 0.020$) and compatibility ($\beta = 0.090$, t = 1.554, $p = 0.002$) had a significant impact on AI adoption. For organizational factors, two factors were analyzed, namely, organizational support ($\beta = -0.025$, t = 0.625, $p = 0.267$), which showed insignificant results, and sustainable human capital ($\beta = 0.223$, t = 5.897, $p = 0.000$), which revealed a significant impact on AI adoption. In terms of environmental factors, both market and customer ($\beta = 0.432$ t = 7.892, $p = 0.000$), and government support ($\beta = 0.080$, t = 1.566, $p = 0.006$) had a significant positive relationship with AI adoption. Hence, H1(b), H1(d), H2(a), H3(a), and H3(b) were supported.

Further, the study found a significant relationship between AI and operational performance ($\beta = 0.557$ t = 14.250, $p = 0.000$), and a relationship between AI and economic performance ($\beta = 0.672$, t = 19.643, $p = 0.000$). Therefore, H4(a) and H4(b) were supported.

## 4.2. Multi-Group Analysis (Firm Size)

The dataset was divided into two groups as per firm size; a multi-group analysis was employed to ascertain the impact of business size on the variables. A study has identified four distinct ways for studying these groups, namely, the parametric approach, permutation approach, confidence-based approach, and Henseler's multi-group approach [117]. In this study, a more advanced extension was introduced, known as the PLS-MGA technique (multi-group analysis) [118]. This approach allows for the identification of significant differences across groups, provided that the observed values fall below 0.05 or exceed 0.95. The authors employed a percentile bootstrapping technique to examine the disparities between the two cohorts of business entities in their research. As mentioned earlier in this study, the authors had 220 responses; 115 responses were from medium-sized firms, and the remaining 105 were from small firms. Therefore, the authors checked the moderating role of firm size in the model (refer to Table 6).

**Table 6.** Multi-group analysis (firm size).

| Moderation Hypotheses | Path (M) | Path (S) | Diff. | PLS MGA (*p*-Value) |
|---|---|---|---|---|
| H5(a) Cost → AI Adoption | 0.122 | 0.080 | 0.042 | 0.763 |
| H5(b) Relative advantage → AI Adoption | 0.087 | −0.162 | 0.250 | 0.048 |
| H5(c) Complexity → AI Adoption | 0.152 | 0.193 | −0.041 | 0.696 |
| H5(d) Compatibility → AI Adoption | −0.021 | 0.197 | −0.218 | 0.122 |
| H6(a) Organizational support → AI Adoption | 0.240 | 0.010 | 0.230 | 0.222 |
| H6(b) Sustainable human capital → AI Adoption | 0.140 | 0.062 | 0.079 | 0.661 |
| H7(a) Market and customer → AI Adoption | −0.161 | 0.084 | −0.245 | 0.169 |
| H7(b) Government support → AI Adoption | 0.247 | 0.296 | −0.048 | 0.785 |
| H8(a) AI adoption → ECP | 0.940 | 0.892 | 0.047 | 0.014 |
| H8(b) AI adoption → OP | 0.922 | 0.894 | 0.028 | 0.141 |

Note: ECP = economic performance; OP = operational Performance; PLS-MGA *p*-value below 5% and above 95% indicates significant values. Diff. = path coefficient differences.

The *p*-value results obtained from the partial least squares–multi-group analysis (PLS-MGA) indicate the presence of statistically significant disparities between medium-sized and small-sized SMEs. The results of our analysis indicate a statistically significant disparity in H1(b) ($p = 0.250 < 0.05$). This suggests that the association between relative advantage and AI adoption exhibits greater strength in medium-sized Saudi SMEs as compared to small-sized Saudi SMEs. Furthermore, our research also revealed a statistically significant disparity in accordance with H4(a) ($p = 0.047 < 0.05$). This finding indicates that the correla-

tion between the adoption of artificial intelligence (AI) and economic performance exhibits greater strength inside medium-sized SMEs as compared to small-sized Saudi SMEs.

## 5. Discussion and Conclusions

In terms of technological factors, this study found that cost had an insignificant effect on AI adoption; the result is inconsistent with past studies [40,119,120]. The results of this study reveal that for Saudi SMEs, the cost of implementation does not seem to be a barrier to adopting AI, which implies that Saudi SMEs have sufficient financial resources to invest in technology processes such as artificial intelligence, machine learning, green manufacturing, design, eco-labeling, and packaging. In addition, Saudi SMEs are also able to invest in capacity-building and training their employees to manage and cope with advanced technologies. Relative advantage has a significant impact on AI adoption; this finding is consistent with past studies [74,121]; the results show that managers in Saudi SMEs perceive AI to be better than the existing or substitute technology. The results also show that the relative advantages of AI increase SMEs' willingness to adopt AI; this means that Saudi SMEs feel that the adoption of AI technology has improved and will improve their reputation and corporate image. Complexity showed a negative but insignificant impact on AI adoption; the findings are in line with the previous literature [119,122,123]. The results suggest that AI technology is inherently complex, so Saudi SMEs are not ready to adopt it. Therefore, complexity may be a fundamental problem in the adoption of AI technology, as technology incorporates and combines heterogeneous computing and machine learning technologies and requires insightful knowledge resources [124]. This could have an impact on the adoption of AI among SMEs in Saudi Arabia. However, managers' perceptions about compatibility showed a significant relationship with AI adoption. The findings show that Saudi Arabian SMEs have believed that AI is not simple or easy to learn, but that it is compatible with their current business activities and the setup of the organization. The reason for this may be that innovation complements current business technologies; the application of AI is not a single event but can be described as a process of knowledge-gathering and integration.

For organizational readiness factors, this study found organizational support has a non-significant relationship with AI adoption; the finding is inconsistent with prior studies [70,125]. The results show that management within Saudi SMEs is not encouraging the adoption of AI. Lack of organizational support for AI is mainly due to high costs, long payback times, difficulties in protecting intellectual property, and high follow-up costs. These challenges prevent companies from supporting AI initiatives from the beginning [126]. Sustainable human capital has a significant impact on AI. This finding is consistent with previous research that confirms that sustainable human capital positively impacts the adoption of AI [127–129]. The significant association found in this study between sustainable human capital and the adoption of AI is likely for several reasons. First, human capital is the most critical resource that contributes significantly to the acceptance of sustainable technologies [130]; this study shows that the Saudi workforce is equipped with skills and knowledge, as significant investment has been made in the development of people. Therefore, it seems that human resource practices and employee readiness are potential plus points for accepting AI and other innovations within firms. The correlation between sustainable human capital and the use of artificial intelligence (AI) is intricate and crucial for firms aiming to utilize AI technology, while upholding their dedication to sustainability. The successful use of artificial intelligence (AI) can be enhanced by having sustainable human capital, which encompasses several aspects such as diversity and inclusion and trained employees. The role of human resources (HR) in cultivating a sustainable and well-prepared workforce for the era of artificial intelligence (AI) is of growing importance, as AI continues to change the future of work.

In terms of environmental readiness, market and customer demand factors have a big effect on innovation adoption in Saudi Arabia. This is in line with previous studies [74,76]. The significant relationship between MC and AI adoption, which can be explained by

customer demand for innovative and cutting-edge products, has increased. As a result, organizations believe that the growth potential of AI is immense and are ready to capture the market and take risks in developing eco-innovative products because of this belief. Government support significantly impacted AI adoption; the findings are in line with the available prior literature [74,95,131]. In previous studies, government policies such as providing monetary incentives, scientific resources, pilot projects, and training programs have been identified as driving factors for SMEs to adopt new technology and green practices [74,131–133]. In the case of Saudi Arabia, government support for SMEs for technology is available through the General Authority of Small and Medium-Sized Enterprises in Saudi Arabia. The Kingdom of Saudi Arabia's General Authority for Small and Medium-Sized Enterprises aims to improve firm performance in environmental protection, rehabilitation, conservation and general improvement, pollution prevention and control, and promoting sustainable development. In addition, the SME Authority thoroughly reviews laws, regulates, removes barriers, and facilitates SMEs and entrepreneurs to market their ideas and products. The authority will also help SMEs develop their skills and networks and provide modern technical assistance to companies in pollution control. They will support SMEs with marketing, help them export their goods and services through e-commerce, and work with international stakeholders. As part of Saudi Vision 2030, the kingdom plans to increase SME investment from its current 20% of GDP to 35% to facilitate their access to finance and encourage financial institutions to increase their current lending from 5% to 20%.

The present study has discovered a significant relationship between the adoption of artificial intelligence (AI) and the economic and operational performance of small and medium-sized enterprises (SMEs) in Saudi Arabia. The findings indicate that the use of artificial intelligence (AI) has the potential to provide favorable results, and hence, the economic and operational performance of SMEs increases simultaneously. The relationship between AI and economic performance was found to be significant; the finding supports past studies [76,134,135]. The findings show that Saudi SMEs have realized that eco-innovation is a key factor in financial performance. It suggests that when SMEs adopt technological innovation in their products and process improvements, it is likely to lead to significant changes in the productivity of their resources. These process improvements could reduce costs and, in turn, lead to better financial results. The relationship between AI and operational performance was found to be significant in parallel with the literature available in the past [97,136]. This finding shows that Saudi SMEs believed that the adoption of AI technologies in production and processes that lead to high efficiency and productivity should improve SMEs' internal processes and manufacturing performance. It is also suggests that environmentally friendly practices help reduce pollution and help SMEs achieve some aspects of their operational objectives (e.g., cost reduction, elimination of liabilities, etc.), which in turn increases competitiveness.

The results show a significant difference in the relationship between relative advantage and AI adoption, AI, and environmental performance. Our study found that, compared with small SMEs, medium-sized SMEs have a more substantial impact of relative advantage on AI adoption in the case of Saudi Arabia. Medium-sized SMEs are considered to adopt AI technology better than their existing technology as compared to small SMEs, because large SMEs have sufficient resources and strong infrastructure. Past studies also mentioned that large companies adopt AI and the latest technology more quickly than small ones [137–139].

The major contribution from the discussion and conclusion of this study lies in unravelling distinct patterns and determinants of artificial intelligence (AI) adoption within the context of small and medium-sized enterprises (SMEs) in Saudi Arabia. First, this study challenges the existing literature by revealing that the cost of implementation has an insignificant effect on AI adoption in Saudi SMEs.

A significant finding is the positive impact of sustainable human capital on AI adoption. This underscores the importance of human resource practices and employee readiness in facilitating the acceptance of AI and other innovations. This study underlines the critical

role of external factors such as market demand and government support in fostering a conducive environment for AI adoption in Saudi SMEs. Further, this study also indicates that the strategic use of AI can lead to simultaneous improvements in economic and operational performance.

One important contribution is that this study found that the relationship between relative advantage and AI adoption is very different depending on the size of the small businesses. Medium-sized SMEs exhibit a more substantial impact, potentially due to their enhanced resources and infrastructure compared to their smaller counterparts.

## 6. Practical, Policy, and Theoretical Implications

The findings of the study indicate that, in contrast to prior research, the economic implications associated with the adoption of artificial intelligence do not prove to be a substantial obstacle for small and medium-sized enterprises (SMEs) in Saudi Arabia. This suggests that these enterprises have sufficient financial capabilities to allocate funds towards the adoption of cutting-edge technologies. As a result, Saudi small and medium-sized enterprises (SMEs) have the ability to strategically distribute their resources towards artificial intelligence (AI), machine learning, and other new procedures, thereby promoting economic expansion and enhancing their competitive edge.

Academically, educators have the opportunity to employ these findings in order to enhance business and technology curricula, providing students with practical knowledge regarding the intricacies of AI implementation within the specific framework of small and medium-sized enterprises (SMEs) in Saudi Arabia. Most of the STEM graduates have limited seed money and are set up as small startups; hence, the TOE framework, along with other technology models, can be discussed in the class with empirical evidence. The examination of the divergent effects of cost and relative benefit on adoption might provide significant pedagogical resources for comprehending the complexity of technology adoption within various organizational contexts.

Policymakers can utilize the findings of this study in order to customize policies that facilitate the adoption of artificial intelligence (AI) among small and medium-sized enterprises (SMEs) in Saudi Arabia. Acknowledging the crucial significance of governmental support, policy measures may concentrate on mitigating obstacles such as high expense, delayed return on investment periods, and problems pertaining to safeguarding intellectual property rights. These endeavors aim to foster the adoption of artificial intelligence initiatives by small and medium-sized enterprises.

This work makes a valuable contribution to the academic community by enhancing our comprehension of the technological context, organizational readiness, and environmental factors that impact the adoption of artificial intelligence (AI) in small and medium-sized enterprises (SMEs) in Saudi Arabia. The comprehensive analysis of the interconnections between cost, relative advantage, complexity, organizational support, human capital, market demand, and government support contributes significantly to the existing body of knowledge.

## 7. Limitations and Future Research Directions

This study faces several limitations. Firstly, in this research, a cross-sectional survey was undertaken to examine and evaluate hypotheses within the specific setting of Saudi Arabia. It is important to note that the outcomes of longitudinal studies conducted in other developing nations may yield dissimilar findings. Furthermore, this survey exclusively focused on small and medium-sized enterprises (SMEs) operating in the manufacturing sector, while service and non-manufacturing SMEs have not been studied to understand better AI adoption and its impact on different performance dimensions. Second, this study justifies the impact of AI on SME performance. The influence of AI on the construction of financial, environmental, and operational performance is confirmed in the SME context. However, more research is needed to examine other factors that influence the sustainable performance of SMEs. It is important to acknowledge that the study sample was exclusively drawn from small and medium-sized enterprises (SMEs). Therefore, it is advisable for

other businesses to exercise caution when attempting to apply these findings, as there may be inherent disparities that could impact the extent to which these variables influence their respective industries. Further, future studies must also include ethical concerns as part of sustainable HR practices when it comes to using AI. Making sure that AI technologies are used and administered responsibly is in line with sustainability principles.

**Author Contributions:** Conceptualization, Y.A.S. and S.B.; methodology, S.B. and Y.A.S.; software, Y.A.S.; validation, S.B.; formal analysis, Y.A.S.; investigation, S.B.; resources, S.B.; data validation, Y.A.S.; writing—original draft preparation, Y.A.S. and S.B.; writing—review and editing, Y.A.S.; visualization, Y.A.S.; supervision, S.B.; project administration Y.A.S.; funding acquisition, S.B. All authors have read and agreed to the published version of the manuscript.

**Funding:** This research work was funded by Institutional Fund Projects under grant no. (IFPDP-242-22). Therefore, the authors gratefully acknowledge technical and financial support from the Ministry of Education and Deanship of Scientific Research (DSR), King Abdulaziz University: IFPDP-242-22.

**Data Availability Statement:** The datasets analyzed during the current study are available from the corresponding author on reasonable request.

**Conflicts of Interest:** The authors declare no conflicts of interest.

## Appendix A

**Table A1.** Questionnaire items and sources.

| Construct | Sub-Constructs | Item Code | Items |
|---|---|---|---|
| Technological Factors | Cost | CTF1 | Implementing an AI-based system is very expensive. |
| | | CTF2 | Implementation of AI reduces operation cost in the long term. |
| | Complexity | XTF1 | Understanding the AI system is difficult. |
| | | XTF2 | Sharing knowledge of the AI based system is difficult. |
| | Compatibility | TFC1 | Integrating an AI system within the company's existing system is easy. |
| | | TFC2 | AI-based machine learning models are compatible with our existing logistics and manufacturing operations. |
| | Relative advantage | RTF1 | AI technology can provide higher economic benefits. |
| | | RTF2 | AI technology can provide better operational performance. |
| Organizational Factors | Sustainable human capital | SHC1 | Our employees are ready with skills necessary for AI implementation. |
| | | SHC2 | Our employees work in teams to implement AI technology. |
| | | SHC3 | Our employees have less knowledge related to machine learning algorithms and AI-based predictive models. |
| | | SHC4 | Our employees receive full support to get training to learn new skills to implement AI technology. |
| | Organizational support | OS1 | Top management encourages employees to learn big data analytics-powered artificial intelligence. |
| | | OS2 | Our company provides rewards for employees that have skills and knowledge related to AI-based systems. |
| | | OS3 | Our company provides resources for employees to learn big data analytics-powered artificial intelligence. |

**Table A1.** *Cont.*

| Construct | Sub-Constructs | Item Code | Items |
|---|---|---|---|
| Environmental Factors | Government support | GS1 | Government provides financial support (subsidies) for adopting AI. |
| | | GS2 | Government provides technical assistance for adopting AI. |
| | | GS3 | Government provides support in training manpower with AI technologies. |
| | | GS4 | A financial credit loan from banks for AI implementation is easy for SMEs. |
| | Market and customer factors | MC1 | Our customers require us to improve our products through AI. |
| | | MC2 | AI can work as an incentive to capture a significant market share for a firm. |
| | | MC3 | Potential gain of publicity and advertising for AI is higher. |
| AI Adoption | | AI1 | My firm is willing to adopt AI to improve firm performance. |
| | | AI2 | My firm adopts new technologies. |
| | | AI3 | My firm cannot make use of AI. |
| Firm Green Performance | Economic | ECP1 | AI will increase our organization's market share. |
| | | ECP2 | AI will increase our organization's corporate profitability. |
| | Operational | OP1 | AI will increase our organization's operational efficiency. |
| | | OP2 | AI will increase our organization's product/service quality. |

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
