# Peer review of "Artificial Intelligence Adoption by SMEs to Achieve Sustainable Business Performance: Application of Technology–Organization–Environment Framework"

_sustainability, doi:10.3390/su16051864_

Round 1

Reviewer 1 Report

Comments and Suggestions for Authors

Dear writers,

I hope this note finds you well. First, thank you for your efforts on the article. Upon reviewing your work, I have some comments for you: .

1. In various sections of the article, including the title, abstract, and main body, you mention "the most influential factors affecting the adoption of artificial intelligence (AI) by SMEs to achieve sustainable business performance in Saudi Arabia by integrating the Technology-Organization-Environment (TOE) framework." After reviewing the TOE framework from primary sources such as the article by Hart O. Awa, Ojiabo Ukoha Ojiabo, and Longlife E. Orokor published in 2017, it appears that you have utilized many constructs from this model with minimal refinement.  My main comment is: Why do you discuss finding influential factors affecting the adoption of AI when, based on the article, you have implemented the TOE framework with only slight modifications to its constructs?

2. The Technology-Organization-Environment (TOE) framework serves as a theoretical foundation for your study. However, I noticed the absence of a literature review that discusses the implementation of this model in prior research. Could you clarify how your work diverges from existing literature? Specifically, what innovative elements have you introduced in your application of the TOE framework?

3-Finally, according to the article by Tornatzky, Louis G.; and Fleischer, Mitchell titled "The Processes of Technological Innovation," the TOE framework serves as a theoretical model that elucidates technology adoption within organizations. It outlines how the adoption and implementation of technological innovations are shaped by the technological context. In light of this, I believe that AI adoption should be considered as part of the overall readiness for technological innovation, rather than as an independent influencing factor.

Best

Comments on the Quality of English Language

needs to be extensive editing

Author Response

We appreciate your thorough review and constructive guidance. These enhancements have undoubtedly strengthened the paper and contribute to its overall quality. Thank you for your time and consideration.

Comment 1. In various sections of the article, including the title, abstract, and main body, you mention "the most influential factors affecting the adoption of artificial intelligence (AI) by SMEs to achieve sustainable business performance in Saudi Arabia by integrating the Technology-Organization-Environment (TOE) framework." After reviewing the TOE framework from primary sources such as the article by Hart O. Awa, Ojiabo Ukoha Ojiabo, and Longlife E. Orokor published in 2017, it appears that you have utilized many constructs from this model with minimal refinement.  My main comment is: Why do you discuss finding influential factors affecting the adoption of AI when, based on the article, you have implemented the TOE framework with only slight modifications to its constructs?

Response: Thank you for your insightful comments and the time you dedicated to reviewing our manuscript. We appreciate the opportunity to address your concerns regarding the utilization of the TOE framework, particularly in relation to the article by Hart O. Awa, Ojiabo Ukoha Ojiabo, and Longlife E. Orokor published in 2017.

We would like to clarify that while we did reference the mentioned paper for the application of the TOE framework in the context of technology adoption, our study incorporates distinct constructs. Our research extends beyond the constructs of Technology, Organization, and Environment utilized in the referenced article. We have tailored the framework to our specific research context, introducing modifications and novel variables such as Cost of implementation (Technological construct), SHR (Organizational construct) and Government Support (Environmental Construct) and these additional constructs align with the unique dynamics of AI adoption in the Saudi Arabian SME context.

Furthermore, we have taken a nuanced approach by introducing Firm Size as a moderator variable in our conceptual model. This deliberate choice reflects our commitment to capturing the intricacies of organizational dynamics in the context of AI adoption. Firm Size is not considered a component of the organizational variable but assumes a distinct role as a moderator, contributing to the richness and depth of our analysis.

We acknowledge the foundational contributions of existing frameworks, including the TOE model, to the field of technology adoption. Our modifications and additions aim to enhance the applicability of the extended TOE framework to the specific challenges and opportunities presented by AI adoption in Saudi Arabian SMEs. We hope this clarification addresses your concerns regarding the uniqueness of our study's conceptual framework.

Comment 2. The Technology-Organization-Environment (TOE) framework serves as a theoretical foundation for your study. However, I noticed the absence of a literature review that discusses the implementation of this model in prior research. Could you clarify how your work diverges from existing literature? Specifically, what innovative elements have you introduced in your application of the TOE framework?

Response: We have added concise and relevant literature review that highlights how Technology-Organization-Environment (TOE) framework has been utilized in prior research and how our work diverges and makes a contribution to implementation and the existing knowledge. We have added the cost of implementation as unique variable. In this study, the authors conceptualize performance as a two-dimensional construct (i.e., operational and economic performance). To the author’s knowledge, this study is the first research work focusing on SMEs AI adoption and its impact on performance as a two-dimensional construct (i.e., operational and economic performance) in Saudi Arabia. Lastly, firm size was used as a moderator variable to understand the group difference between small and medium-sized SMEs.

Comment 3. Finally, according to the article by Tornatzky, Louis G.; and Fleischer, Mitchell titled "The Processes of Technological Innovation," the TOE framework serves as a theoretical model that elucidates technology adoption within organizations. It outlines how the adoption and implementation of technological innovations are shaped by the technological context. In light of this, I believe that AI adoption should be considered as part of the overall readiness for technological innovation, rather than as an independent influencing factor.

Response:

The T-O-E framework provides a comprehensive framework for analysing the interplay between technology characteristics, organizational factors, and environmental influences in shaping the adoption decisions of organizations. Technological context includes the characteristics related to technology being implemented or its adoption decision. Many studies have kept technology adoptions such as Machine learning, Cloud Computing, Artificial Intelligence, Green Banking, E-health technology as dependent variable in the TOE framework and also as independent influencing variable to performance. Many studies have employed it to explain the adoption of specific technological innovations. For instance, the model has been applied to understand the adoption of  electronic customer relationship management (eCRM) systems (Racherla & Hu, 2008), cloud computing (Low, Chen and Wu, 2011; Oliveira, Thomas, & Espadanal, 2014), geographical information technologies (Amade, Oliveira, & Painho, 2020), blockchain technology (Malik et al., 2021; Ganguly, 2022), e-business (Satar & Alarifi, 2022), green banking practice (Aslam & Jawaid, 2023), Machine Learning (Das & Bala, 2023) and other emerging technologies.

  1. Amade, N., Oliveira, T., & Painho, M. (2020). Understanding the determinants of GIT post-adoption: perspectives from Mozambican institutions. Heliyon6(5).
  2. Aslam, W., & Jawaid, S. T. (2023). Green banking adoption practices: the pathway of meeting sustainable goals. Environment, Development and Sustainability, 1-26.
  3. Das, S. D., & Bala, P. K. (2023). What drives MLOps adoption? An analysis using the TOE framework. Journal of Decision Systems, 1-37.
  4. Ganguly, K. K. (2022). Understanding the challenges of the adoption of blockchain technology in the logistics sector: the TOE framework. Technology Analysis & Strategic Management, 1-15.
  5. Low, C., Chen, Y., & Wu, M. (2011). Understanding the determinants of cloud computing adoption. Industrial management & data systems111(7), 1006-1023.
  6. Malik, S., Chadhar, M., Vatanasakdakul, S., & Chetty, M. (2021). Factors affecting the organizational adoption of blockchain technology: Extending the technology–organization–environment (TOE) framework in the Australian context. Sustainability13(16), 9404.
  7. Oliveira, T., Thomas, M., & Espadanal, M. (2014). Assessing the determinants of cloud computing adoption: An analysis of the manufacturing and services sectors. Information & management51(5), 497-510.
  8. Racherla, P., & Hu, C. (2008). eCRM system adoption by hospitality organizations: A technology-organization-environment (TOE) framework. Journal of Hospitality & Leisure Marketing17(1-2), 30-58.
  9. Satar, M. S., & Alarifi, G. (2022). Factors of E-business adoption in small and medium enterprises: evidence from Saudi Arabia. Human Behavior and Emerging Technologies2022.

More over we have made substantial additions to the methodology section to provide a more comprehensive understanding of the research design. Specifically, we have addressed the method of identifying prospective respondents, instrument design, target audience, and the time frame for data collection. The revised methodology section now reads as follows:

“The self-administered survey method with random sampling was adopted to obtain data for this study. Random sampling is considered the most suitable strategy due to the equal probability assigned to each unit (Secker, 1995). Prospective respondents were identified through a systematic approach, leveraging industry directories, business associations, and government records. The finalized questionnaire was distributed to SME executives at middle or senior level or the owner/entrepreneur from the construction, energy, logistics, manufacturing, and services industries in the Jeddah industrial area from March 2023 till May 2023. Approximately 300 respondents were approached and 220 valid responses from the industry were processed for data analysis”.

Laslty, we have incorporated explicit implications in the manuscript, addressing various dimensions such as economic impact, teaching applications, policy influence, societal impact, and contributions to the body of knowledge. The practical and theoretical implications are now presented separately in dedicated subsection, and you may check section 6 in the manuscript.

Reviewer 2 Report

Comments and Suggestions for Authors

1.      The contribution of the paper is not clear. The authors should provide an explicit comparison between the existing studies and the current paper, and should explain what drives the differences in the results, not just the differences in the results. Each paragraph should be in standard style because it is easier for the reviewer to read. I discovered that certain sections in the paper are excessively long.

2.      The literature section is not well-updated. Neither shows the gap, portraying the research progress on the relevant topic and even looking incomplete. The author needs to rewrite this section. I also noticed a weak conceptual development. This section of the conceptual development does not provide a sophisticated discussion of the concept of AI. The existing content is very much centered on the descriptive background of AI. The authors should develop the literature section more fully and specifically.

3.      Where is the moderation hypothesis? Please provide relevant conceptual definitions for all variables.

4.      The methodology section is impoverished:" Please explain/describe the method of identifying prospective respondents. How did the researchers identify who to send which channel used, sent, and returned?  Although this study used enough sample size. However, the methodology section is under-presented. How about instrument design? Target audience? Time frame?

5.      Why Saudi Arabia was selected for conducting this research? In addition, more importantly, how does it contribute to investigating the research question? What are the implications for other developing and developed countries?

6.      The process for selecting the items to construct the measurement scales is unclear. The authors should explain the steps one by one.

7.      The authors used PLS-SEM instead of CB-SEM, so they have to provide justification for why they preferred PLS-SEM over CB-SEM.

8.      How can the research be used in practice (economic and commercial impact), in teaching, to influence public policy, and in research (contributing to the body of knowledge)?  What is the impact upon society (influencing public attitudes, affecting quality of life)?  Are these implications consistent with the findings and conclusions of the paper? After incorporating the above comments, all theoretical and managerial contributions can be explained accordingly.

9.      Where is the practical and theoretical implication of the research? Authors should add implications in the subsection separately. 

10.  The quality of communication is OK I enjoyed reading it, but random language errors are visible, like page # 6, line # 275 and so on. Studying AI's mechanisms and important determinants “on” business performance is theoretically and practically valuable (Chen and Lin, 2021).

Comments on the Quality of English Language

The authors are suggested to proofread the whole paper enhancing the quality of English.

Author Response

Comment 1: The contribution of the paper is not clear. The authors should provide an explicit comparison between the existing studies and the current paper, and should explain what drives the differences in the results, not just the differences in the results. Each paragraph should be in standard style because it is easier for the reviewer to read. I discovered that certain sections in the paper are excessively long.

Response: Thank you for your valuable feedback. We appreciate your insightful comments on the clarity of the paper's contribution.

In response to your suggestion, we have enhanced the manuscript by providing a more explicit comparison with existing studies in the discussion section. We have highlighted the specific factors driving differences in results, moving beyond the mere presentation of disparities.

Furthermore, we have revised the structure to adhere to standard paragraph styles for improved readability. We acknowledge that certain sections may be excessively long, and we have addressed this issue by appropriately subdividing content for better organization and clarity.

Comment 2: The literature section is not well-updated. Neither shows the gap, portraying the research progress on the relevant topic and even looking incomplete. The author needs to rewrite this section. I also noticed a weak conceptual development. This section of the conceptual development does not provide a sophisticated discussion of the concept of AI. The existing content is very much centered on the descriptive background of AI. The authors should develop the literature section more fully and specifically.

Response: We acknowledge the concerns raised regarding the literature section. In response, we have made significant revision and added the literature review, addressing gaps, showcasing research progress, and providing a more sophisticated discussion of the AI adoption research with TOE framework integrated. Kindly refer to manuscript section 2.1.

Comment 3: Where is the moderation hypothesis? Please provide relevant conceptual definitions for all variables.

Response: We appreciate your careful examination of our manuscript. We would like to clarify that the moderation hypothesis is detailed in Section 2.3.5 under the heading "Firm Size as a Moderator." In this section, we explicitly outline our hypothesis, stating that the association between relative advantage and AI adoption is anticipated to exhibit greater strength in medium-sized Saudi SMEs compared to small-sized ones. And all other hypotheses are presented.

Regarding your comment on conceptual definitions, we would like to draw your attention to the introductory portion of each variable, where we have provided clear and concise conceptual definitions at the beginning of their respective sections. We believe these definitions offer a comprehensive understanding of the constructs under investigation.

Should you require any further clarification or have additional suggestions, please feel free to let us know. We are committed to addressing your comments and enhancing the quality of our manuscript.

Comment 4: The methodology section is impoverished:" Please explain/describe the method of identifying prospective respondents. How did the researchers identify who to send which channel used, sent, and returned?Although this study used enough sample size. However, the methodology section is under-presented. How about instrument design? Target audience? Time frame?

Response: We have carefully considered your comments and have made substantial additions to the methodology section to provide a more comprehensive understanding of the research design. Specifically, we have addressed the method of identifying prospective respondents, instrument design, target audience, and the time frame for data collection. The revised methodology section now reads as follows:

“The self-administered survey method with random sampling was adopted to obtain data for this study. Random sampling is considered the most suitable strategy due to the equal probability assigned to each unit (Secker, 1995). Prospective respondents were identified through a systematic approach, leveraging industry directories, business associations, and government records. The finalized questionnaire was distributed to SME executives at middle or senior level or the owner/entrepreneur from the construction, energy, logistics, manufacturing, and services industries in the Jeddah industrial area from March 2023 till May 2023. Approximately 300 respondents were approached and 220 valid responses from the industry were processed for data analysis”.

Comment 5: Why Saudi Arabia was selected for conducting this research? In addition, more importantly, how does it contribute to investigating the research question? What are the implications for other developing and developed countries?

Response: The selection of Kingdom of Saudi Arabia as the research context for this study was purposeful and driven by several considerations that align with the research question and contribute meaningfully to the broader understanding of technology adoption. Currently, KSA is implementing rapid modernization initiatives to achieve the ambitious Saudi Vision 2030. This dynamic environment provides an intriguing context for investigating the factors influencing the adoption of artificial intelligence (AI) within the specific context of Small and Medium-sized Enterprises (SMEs). Moreover, the economy of Saudi Arabia is one of the top twenty economies in the world, and the largest economy in the Arab world and the Middle East North Africa (MENA region). Saudi Arabia is also part of the G20 group of countries. Therefore, the selection of Saudi Arabia as the research setting was deliberate. The findings are expected to hold implications not only for Saudi SMEs but also for SMEs in other developing and developed countries, offering valuable insights for policymakers, researchers, and practitioners alike. 

Concise version of above explanation has been added in the methodology section.

Comment 6: The process for selecting the items to construct the measurement scales is unclear. The authors should explain the steps one by one.

Response: The questionnaire was meticulously designed, incorporating the most significant influencing factors identified in prior research. Here is a step-by-step explanation of the process:

  1. Identification of Constructs:
    • The initial step involved identifying the key constructs based on a thorough review of relevant literature on innovation adoption. Constructs were selected to comprehensively cover the technological, organizational, and environmental factors influencing AI adoption.
  2. Adaptation of Constructs:
    • The selected constructs were adapted and adjusted to align with the specific requirements of our research in the context of Saudi SMEs. This process ensured that the measurement scales were tailored to the unique characteristics of the target population.
  3. Technology Factor:
    • The technology factor encompassed four sub-constructs—cost, relative advantage, complexity, and compatibility. These sub-constructs were derived from established studies (Lin & Ho, 2011b; Sahin, 2006; Sia et al., 2016; Awa Hart & Ojiabo, 2016; Beatty et al., 2001).
  4. Organization Factor:
    • The organization factor included two sub-constructs—sustainable human capital and organizational support. These sub-constructs were selected based on their relevance to the adoption of AI technology (Aboelmaged & Hashem, 2019; Chang & Chen, 2012; Jun et al., 2019).
  5. Environmental Factor:
    • The environmental factor was measured by two sub-constructs—government support and market and customer factors. These sub-constructs were adapted from a study by Jun et al. (2019).
  6. AI Adoption (AIA) Items:
    • Items related to AI adoption (AIA) were modified in the questionnaire and adopted from previous studies to ensure relevance and alignment with the research context (Aboelmaged & Hashem, 2019; Kusi-Sarpong et al., 2015).
  7. Sustainable Business Performance:
    • Sustainable business performance consisted of two distinct components—economic performance and operational performance. These measures were modified and adapted from prior research to capture the specific outcomes relevant to our study (Baeshen et al., 2021; Zhang et al., 2020).

Comment 7: The authors used PLS-SEM instead of CB-SEM, so they have to provide justification for why they preferred PLS-SEM over CB-SEM.

Response: We would like to provide our justification for choosing PLS-SEM instead of CB-SEM. PLS-SEM is generally more flexible and well-suited for complex models with many latent variables and indicators. As our model is complex, intricate and includes formative constructs, PLS-SEM was a more appropriate choice.

The present study utilizes the Technology-Organization-Environment (TOE) framework to construct a research model that explains the readiness of firms towards the adoption of AI and the firm performance within SMEs in emerging countries like Saudi Arabia. Therefore, our primary goal is prediction rather than testing theory, PLS-SEM is often preferred over CB-SEM. Further, PLS-SEM is often recommended for exploratory research where the goal is to discover relationships among variables rather than confirming a pre-specified theory.

Moreover, our independent variables are having effect on predictor variable, hence endogeneity is a significant concern in our research, PLS-SEM is considered more flexible regarding endogeneity assumptions compared to CB-SEM.

Lastly, there is high multicollinearity in the outer model, this can be seen in figure 2, kindly check the observed variables and latent variables. PLS-SEM is less affected by multicollinearity among independent variables. With high multicollinearity in the model, PLS-SEM is more appropriate choice.

Comment 8: How can the research be used in practice (economic and commercial impact), in teaching, to influence public policy, and in research (contributing to the body of knowledge)?  What is the impact upon society (influencing public attitudes, affecting quality of life)?  Are these implications consistent with the findings and conclusions of the paper? After incorporating the above comments, all theoretical and managerial contributions can be explained accordingly.

Comment 9: Where is the practical and theoretical implication of the research? Authors should add implications in the subsection separately.

Response to comment 8 & 9 combined:

Thank you for your thoughtful comments and recommendations. We have carefully considered your feedback and incorporated explicit implications in the manuscript, addressing various dimensions such as economic impact, teaching applications, policy influence, societal impact, and contributions to the body of knowledge. The practical and theoretical implications are now presented separately in dedicated subsection, and you may check section 6 in the manuscript.

Comment 10: The quality of communication is OK I enjoyed reading it, but random language errors are visible, like page # 6, line # 275 and so on. Studying AI's mechanisms and important determinants “on” business performance is theoretically and practically valuable (Chen and Lin, 2021).

Response:

We have checked again and thoroughly reviewed the manuscript to remove any language or grammar errors. We appreciate your thorough review and constructive guidance. These enhancements have undoubtedly strengthened the paper and contribute to its overall quality. Thank you for your time and consideration.

Round 2

Reviewer 1 Report

Comments and Suggestions for Authors
  • Title vs. Abstract Discrepancy: The title of your paper references "antecedents," but the abstract immediately shifts focus to "investigating and proposing a theoretical model." These appear to be two distinct objectives. If you are applying the Technology-Organization-Environment (TOE) framework in your research, especially within the context of SMEs, it is crucial to clarify whether your primary aim is to identify antecedents of AI adoption or to develop a theoretical model. The research design for each of these objectives would inherently differ.
  • Research Method Clarification: In the research methodology section (line 417), you mention interviewing five top managers, but the purpose and relevance of these interviews are not clear. How do the findings from these interviews align with your main research objective of identifying AI adoption antecedents? There seems to be a disconnect between the results of these interviews and the primary research findings.
  • Multi-group Analysis Consistency: In Section 4.1, "Multi-group Analysis (Firm Size)," you investigate firm size by posing questions to managers of both small and medium firms. However, if you acknowledge that the sample groups (small and medium firms) are inherently different, it raises questions about the validity of the results in the initial sections of your research. You initially suggest that your sample is homogeneous and shares similar characteristics. This inconsistency might affect the reliability of your findings and should be addressed for a clearer understanding of the research implications.
Comments on the Quality of English Language

Extensive editing of English language required

Author Response

Thank you for your insightful and constructive feedback on our manuscript. We appreciate the time and effort you have invested in reviewing our work again, and we have carefully considered your comments to enhance the quality of our research. Below, we provide detailed responses to the key points you raised in the second round:

Comment 1- Title vs. Abstract Discrepancy: The title of your paper references "antecedents," but the abstract immediately shifts focus to "investigating and proposing a theoretical model." These appear to be two distinct objectives. If you are applying the Technology-Organization-Environment (TOE) framework in your research, especially within the context of SMEs, it is crucial to clarify whether your primary aim is to identify antecedents of AI adoption or to develop a theoretical model. The research design for each of these objectives would inherently differ.

Reply 1. Title vs. Abstract Discrepancy:

We acknowledge the concern regarding the apparent discrepancy between the title and abstract of our paper. To address this, we have revised both the title and abstract to ensure they accurately reflect the focus and objectives of our study. Now you will find that our primary aim is to investigate which factors of TOE framework influence the AI adoption within the context of SMEs. We have clarified the language to eliminate any ambiguity and better align the title with the abstract and objectives (Check line 78 to 83).

Comment 2- Research Method Clarification: In the research methodology section (line 417), you mention interviewing five top managers, but the purpose and relevance of these interviews are not clear. How do the findings from these interviews align with your main research objective of identifying AI adoption antecedents? There seems to be a disconnect between the results of these interviews and the primary research findings.

Reply 2. Research Method Clarification:

We appreciate your observation regarding the lack of clarity on the purpose and relevance of the interviews with top managers. In the revised manuscript, we have provided a more explicit explanation of how these interviews contribute to our overall research variable selection from the TOE framework. By doing so, we aim to establish a clearer link between the interview findings and the primary research outcomes. We have added further explanation in the methodology section:

“To finalize the questionnaire, the initial step involved identifying the key constructs based on a thorough review of relevant literature on innovation adoption. Constructs were selected to comprehensively cover the technological, organizational, and environmental factors influencing AI adoption. Previous studies reveal that the TOE framework encompasses a multitude of variables. Hence, the interviews with five managers were conducted to identify and prioritize the variables within the Technology-Organization-Environment (TOE) framework that are most relevant to the context of Saudi Arabian SMEs. After the interviews, constructs were finalized for the study.”

Comment 3-Multi-group Analysis Consistency: In Section 4.1, "Multi-group Analysis (Firm Size)," you investigate firm size by posing questions to managers of both small and medium firms. However, if you acknowledge that the sample groups (small and medium firms) are inherently different, it raises questions about the validity of the results in the initial sections of your research. You initially suggest that your sample is homogeneous and shares similar characteristics. This inconsistency might affect the reliability of your findings and should be addressed for a clearer understanding of the research implications.

Reply 3. Multi-group Analysis Consistency:

Your feedback regarding the potential inconsistency in our approach to investigating firm size.

Our data analysis approach:

First, we created a structural model that represents the relationships between variables in our study. The structural equation modeling (SEM) analysis was performed on the entire data to find the structural relationships or parameter estimates in a model. We reported and explained all the relationships.

Second step in our approach was to compare the structural relationships or parameter estimates in a model across two distinct groups. It is commonly applied to examine whether certain relationships or patterns hold consistently across different subgroups. This involves estimating parameters and assessing the model fit within each group. We have clearly reported the results of the multigroup analysis that sample size reduced as we divided into two groups, moreover we have included any observed differences and the statistical significance of those differences.

Multigroup analysis was an additional statistical test to show that there is difference in the results so in future researcher may considered this firm size factor as significant variable in SME study.

Previous study that has utilized same approach Multigroup analysis:

Lutfi, A., Al-Okaily, M., Alsyouf, A., Alsaad, A., & Taamneh, A. (2020). The impact of AIS usage on AIS effectiveness among Jordanian SMEs: A multi-group analysis of the role of firm size. Global Business Review, 0972150920965079.

Thank you once again for your valuable feedback.

Best regards

Round 3

Reviewer 1 Report

Comments and Suggestions for Authors

Comment 1: Your sentence 420-224, 'Previous studies reveal that the TOE framework encompasses a multitude of variables. Hence, the interviews with five managers were conducted to identify and prioritize the variables within the Technology-Organization-Environment (TOE) framework that are most relevant to the context of Saudi Arabian SMEs. After the interviews, constructs were finalized for the study,' could be improved as follows:

'In the context of our research, we customized the TOE framework originally proposed by Tornatzky & Fleischer (1990), as elaborated in Section 2. Our approach involved selecting the variables within the Technology-Organization-Environment (TOE) framework that are most pertinent to Saudi Arabian SMEs. This selection process was informed by interviews with five managers.

Comment  2: In agreement with the improved sentence above, it would be beneficial to explicitly list and discuss the variables studied within the literature section. By doing so, you can provide a comprehensive overview of the relevant concepts.

Furthermore, consider dedicating a separate section to present the results obtained from the interviews with managers. This section can serve as the bridge between the literature review and the development of the conceptual model and hypothesis. By presenting the feedback and insights gathered from these interviews, you can establish a strong foundation for your research, ultimately leading to the formulation of a robust conceptual model and well-defined hypotheses.

Additionally, it's important to address the issue of the interview results and the role of the interviewer in shaping the conceptual model. Ensure that the link between the interview data and the development of the conceptual model is clear and emphasized in your paper. This will help readers understand the significance of the interviews in informing your research.

Comments on the Quality of English Language

Extensive editing of English language required

Author Response

Comment 1:

Reply: We appreciate your feedback, and we agree that the revised wording you provided improves the articulation of our methodology.

In light of your recommendation, we have revised the sentence (420-424) as follows:

"In the context of our research, we customized the TOE framework originally proposed by Tornatzky & Fleischer (1990), as elaborated in Section 2. Our approach involved selecting the variables within the Technology-Organization-Environment (TOE) framework that are most pertinent to Saudi Arabian SMEs. This selection process was informed by interviews with five managers."

We believe that this modification aligns more effectively with your suggested improvements while maintaining the accuracy and coherence of our methodology.

Comment2:

Reply 2:

Regarding the incorporation of a dedicated section for presenting the results obtained from the interviews with managers. We appreciate your thoughtful suggestion to establish a connection between the literature review and the development of the conceptual model and hypotheses through this additional section. We completely agree with the importance of showcasing the feedback and insights gathered from the interviews as a foundation for our research.

We have added a paragraph seamlessly continuing from the previous content, which includes a summary of the results from the interviews and the applied selection criteria. Kindly review the recently incorporated Table 1.
